# Placental Transcription Profiling in 6–23 Weeks’ Gestation Reveals Differential Transcript Usage in Early Development

**DOI:** 10.3390/ijms23094506

**Published:** 2022-04-19

**Authors:** Konstantinos J. Bogias, Stephen M. Pederson, Shalem Leemaqz, Melanie D. Smith, Dale McAninch, Tanja Jankovic-Karasoulos, Dylan McCullough, Qianhui Wan, Tina Bianco-Miotto, James Breen, Claire T. Roberts

**Affiliations:** 1Adelaide Medical School, University of Adelaide, Adelaide, SA 5005, Australia; konstantinos.bogias@adelaide.edu.au (K.J.B.); shalem.leemaqz@flinders.edu.au (S.L.); dale.mcaninch@gmail.com (D.M.); tanja.jankovickarasoulos@flinders.edu.au (T.J.-K.); 2Robinson Research Institute, University of Adelaide, Adelaide, SA 5005, Australia; tina.bianco@adelaide.edu.au; 3Dame Roma Mitchell Cancer Research Laboratories, Adelaide Medical School, University of Adelaide, Adelaide, SA 5005, Australia; stephen.pederson@adelaide.edu.au; 4Flinders Health and Medical Research Institute, Flinders University, Bedford Park, SA 5042, Australia; melanie.smith@flinders.edu.au (M.D.S.); dylan.mccullough@flinders.edu.au (D.M.); qianhuiwan@chosenmedtech.com (Q.W.); 5School of Agriculture, Food and Wine, Waite Research Institute, University of Adelaide, Adelaide, SA 5005, Australia; 6Indigenous Genomics, Telethon Kids Institute (Adelaide Office), Adelaide, SA 5000, Australia; jimmy.breen@telethonkids.org.au; 7College of Health & Medicine, Australian National University, Canberra, ACT 2600, Australia

**Keywords:** RNA-seq, human, placenta, development, transcriptome

## Abstract

The human placenta is a rapidly developing transient organ that is key to pregnancy success. Early development of the conceptus occurs in a low oxygen environment before oxygenated maternal blood begins to flow into the placenta at ~10–12 weeks’ gestation. This process is likely to substantially affect overall placental gene expression. Transcript variability underlying gene expression has yet to be profiled. In this study, accurate transcript expression profiles were identified for 84 human placental chorionic villus tissue samples collected across 6–23 weeks’ gestation. Differential gene expression (DGE), differential transcript expression (DTE) and differential transcript usage (DTU) between 6–10 weeks’ and 11–23 weeks’ gestation groups were assessed. In total, 229 genes had significant DTE yet no significant DGE. Integration of DGE and DTE analyses found that differential expression patterns of individual transcripts were commonly masked upon aggregation to the gene-level. Of the 611 genes that exhibited DTU, 534 had no significant DGE or DTE. The four most significant DTU genes *ADAM10*, *VMP1*, *GPR126*, and *ASAH1,* were associated with hypoxia-responsive pathways. Transcript usage is a likely regulatory mechanism in early placentation. Identification of functional roles will facilitate new insight in understanding the origins of pregnancy complications.

## 1. Introduction

The placenta undergoes rapid growth and development across a short lifespan during gestation [1]. The myriad functions of the placenta include transport of nutrients, gases, and wastes between the maternal and fetal circulations, mediation of maternal immune tolerance, regulation of maternal insulin sensitivity and protection of the fetus against xenobiotics [2]. Successful initiation of placentation involves trophoblast differentiation along both villus and extravillous pathways. Extravillous cytotrophoblasts (EVTs) form a cytotrophoblastic shell in the first few weeks’ post conception which completely encircles the conceptus. EVTs invade, colonize, and occlude the uteroplacental arteries during the first trimester of development [3]. Occluding EVTs in the spiral arterioles, so-called trophoblast plugs, begin to be dislodged at around 10–12 weeks’ gestation, facilitating the flow of oxygen-rich maternal blood into the placental intervillous space [1,4]. Thus, there is a transition between two biologically distinct developmental environments over early to mid gestation, with a physiological low oxygen environment up to ~10 weeks’ gestation, and a physiologically “normal” oxygen environment developing after this time. The shift from a low to an increasingly normal physiological oxygen environment late in the first trimester [4] is a critical time in placental development, with deficiency in EVT invasion associated with complications ranging from miscarriage to preeclampsia [5,6]. Transcriptional dynamics across gestation have previously been profiled in gene expression studies, including between mid gestation and term [7], first and third trimester [8,9] and more recently between 6–10 weeks’ and 11–23 weeks’ gestation [10]. Significant changes in microRNAs (miRNAs) have also been reported between the first and third trimesters [11] and between 6–10 weeks’ and 11–23 weeks’ gestation with placenta-specific miRNA clusters reflected in maternal plasma [12]. Roles of long non-coding RNA (lncRNA) in placental development have also been reviewed including their role as *cis* regulators of autosomal gene expression [13], and in pregnancy complications and immune system regulation [14].

Approximately 95% of all multi-exon genes have more than one alternatively spliced form which serves to increase protein diversity [15,16]. Profiling at the transcript-level enables detection of differential transcript expression (DTE) within a gene that may be masked upon aggregation to the gene-level [17]. Most previous studies have used differential gene expression (DGE) analysis to characterize the placental transcriptome, despite evidence that individual transcript isoforms perform their own distinct functions in human placenta [18,19,20,21]. While differential expression analysis is an indispensable tool to comprehensively profile transcriptional dynamics, it is limited in that it ignores underlying changes in transcript usage. Indeed, the presence of variable isoform usage may hinder detection of significant changes in gene expression [22]. Differential transcript usage (DTU) analysis captures the changes in transcript proportions which can uncover the contribution of individual transcripts to overall gene expression [23,24]. To date, relatively few studies on placenta have investigated the impact of transcript usage on the placental transcriptome and these have been primarily focused on preeclampsia and intrauterine growth restriction (IUGR) [25,26] and EVT differentiation [27]. Assessing DTU in early gestation may offer a unique perspective on the regulation of placental differentiation and growth. Changes in protein abundance altered by isoform usage and intron retention have been previously identified in human breast adenocarcinoma cells using RNA-seq integrated with mass spectrometry [28], highlighting the functional significance of DTU. In placenta, several genes have already been shown in previous studies to exhibit alternative expression of transcript isoforms throughout development including *PlGF* [19,29], *VEGF* [30], *IGF2* [20], and *FLT1* [21,31], suggesting modulation of gene expression through usage of specific transcripts.

Profiling individual transcript expression and DTU across 6–23 weeks’ gestation can provide novel insights on the effects of varying oxygen tension during this dynamically changing period in placental development. At present, the complexities of variable transcript usage in human placenta across early gestation remain understudied, attributable to the paucity of samples reported in the literature and compounded by the difficulties in sampling the placenta during ongoing pregnancy. In order to investigate the effects of variable transcript expression and usage, a previously studied gene expression dataset of 84 chorionic villus tissue samples (6–23 weeks’ gestation) [10] was used to identify transcript expression profiles. In this study DGE, DTE and DTU analyses were performed to identify changes in the expression and usage of individual transcripts with respect to gene expression. Gene ontology (GO) enrichment analysis of significant genes highlighted enrichment of biological and molecular processes driven by DGE, DTE, and DTU. Profiling at a resolution of individual transcripts allowed identification of potentially functional impacts resulting from transcript variability in samples between 6–10 weeks’ and 11–23 weeks’ gestation.

## 2. Results

After initial filtering of the data, a total of 14,687 transcripts were detected. These belonged to 10,078 unique genes, of which 3028 had more than one detectable isoform. Of the 14,687 transcripts, the 10 with the highest expression across all 84 samples, in order of expression normalized by transcript length, were *MTRNR2L12-001*, *MTRNRL2-001*, *CGA-001*, *MTRNR2L8-001*, *CSH1-001*, *MTRNR2L1-001*, *CSH2-001*, *WDR74-005*, *KISS1-001*, and *PSG3-001* (Appendix A). Principal component analysis of transcript expression showed that 26.22% of variance within the data was explained by PC1 indicating that gestational age represents the largest source of variability within the data (Figure 1).

### 2.1. Changes in the Placental Transcriptome from Early to Mid Gestation Are Highly Enriched for Genes Involved in Cell Migration and Transmembrane Signaling

A total of 1642 genes were significantly different in at least one of three analyses, DGE, DTE and/or DTU. Comparison between analyses revealed the number of genes common to the three approaches (Figure 2). DTE analysis identified a total of 1011 differentially expressed (DE) transcripts, from 861 genes (Appendix A); DGE analysis identified 879 DE genes; and 611 genes showed observable changes in transcript usage (DTU; FDR < 0.05). There were also 682 genes common to more than one analysis with 27 genes observed in all three (Figure 2). Of the 1642 genes identified across all analyses, 10 were long non-coding RNA (lncRNA). The 10 lncRNAs identified were *AC079612.1*, *AC080112.2*, *C8orf31*, and *LINC01554* in DGE; *MSC-AS1*, *LINC02860*, *C1QTNF1-AS1* and *MIRLET7BHG* in both DGE and DTE; *AC110619.1* in DTU; and *LINC01118* identified in DGE, DTE, and DTU analyses. A total of 2358 transcripts, from the 1395 genes were significant in DTE and DTU analyses. Of the 1395 genes, 632 had one detectable transcript after filtering for low read counts, while 461 genes expressed two transcripts, and 302 expressed three or more transcripts. Of the 2358 transcripts, 1987 were protein coding and 367 were non-coding. The three most abundant types of non-coding transcripts consisted of processed transcripts with no open reading frame (ORF) (*n* = 139), transcripts containing retained introns (*n* = 132), and transcripts targeted by nonsense-mediated decay (NMD) (*n* = 86). Two protein coding transcripts were from immunoglobulin genes and two unprocessed pseudogenes were also in the set.

Changing expression patterns of *PGF* isoforms, previously shown to associate with preeclampsia [29], identified six isoforms of *PGF* (Appendix A), three of which were protein coding and three of which that were non-coding with retained introns. *IGF2* exhibited dominant expression of a single transcript (*IGF2-003*) along with four other minor transcripts, to make a total of five isoforms (Appendix A). The *FLT1* gene contains 30 exons with 5 isoforms (9 isoforms in GRCh38) and has also been shown to be involved in preeclampsia [31]. Three transcripts (*FLT1-001*, *FLT1-201, FLT1-203*) were identified in placenta across 6–23 weeks’ gestation, with dominant expression of *FLT1-203* (Appendix A). Two isoforms of *VEGFB* were also identified (*VEGFB-001*, *VEGFB-005*) in placenta during early to mid-gestation.

GO enrichment analysis of the 1642 significant genes identified Biological Processes such as cellular responses to cell migration, cellular responses to chemical stimulus and organic substances, and defense response. The top Molecular Functions included transmembrane signaling receptor activity, glycosaminoglycan binding, and receptor ligand activity, with cellular components enriched within the plasma membrane (Table 1). As a change in oxygen tension occurs from about 10 weeks’ gestation, significantly enriched GO terms (FDR > 0.05) were queried for any processes relating to oxygen tension or hypoxia response. Terms containing the expressions “oxygen” and “hypoxia” were assessed returning two biological processes which were response to oxygen-containing compound and reactive oxygen species metabolic process. GO enrichment was then performed for each separate analysis and the top biological processes and molecular functions identified. The top enriched terms in DGE were metal ion transmembrane transporter activity, gated channel activity and adaptive immune response. In DTE, they were cell surface receptor signaling pathway, cell communication, and multicellular organismal process. For DTU, top enriched terms were organic anion transport, signaling receptor activator activity, and active transmembrane transporter activity.

Of the 27 genes overlapping all three analyses (Appendix A), the largest number of transcripts were found in *FN1* (*n* = 9), *CD36* (*n* = 6) and *FOS* (*n* = 5). Multiple constituent transcripts with DTE were identified for 17 genes, with *ADGRD1* (formerly *GPR133*), *ANGPTL1*, *GPRC5C*, and *CD36* expressing the greatest numbers of changing transcripts. Of the 27 genes, 25 expressed at least two transcripts with changing proportions between 6–10 weeks’ and 11–23 weeks’ gestation. Both *CD36* and *NRP2*, which had significant DTU at the gene-level, had only one transcript each (*CD36-016*, *NRP2-005*) with significantly changing proportions (Appendix A). Of note, *CD36* had six transcripts that were all statistically significant in DTE (FDR < 0.05, |logFC| > 1), with one transcript (*CD36-016*) statistically significant for changing proportions in transcript-level DTU (FDR < 0.05; Appendix A). One lncRNA (*LINC01118*) was statistically significant in all three analyses and had detectable expression for three isoforms, two of which were significant in DGE, DTE and DTU analysis (*LINC01118-001*, *LINC01118-002*). Apart from one lncRNA, all other genes in the set were protein coding. However, transcript level annotations revealed that five genes expressed transcripts with a retained intron (*ANKRD33*, *CD36*, *FN1*, *GPRC5C*, *PEAK1*), five expressed processed transcripts with no ORF (*CD36*, *F3*, *FOS*, *SBSPON*, *SLC30A2*) and one gene (*HPGD*) expressed a transcript targeted by NMD. Protein coding transcripts were the vast majority, comprising 46 of the 52 transcripts expressed from the set of 27 genes.

### 2.2. Variable Transcript Expression Masks Differential Gene Expression

A greater number of DE transcripts were found to be more highly expressed in 11–23 weeks’ (upregulated) compared to transcripts more highly expressed in 6–10 weeks’ gestation (down-regulated), with 795 transcripts upregulated and 216 transcripts down-regulated, indicating a possible increase in transcriptome complexity with increasing differentiation of the developing tissue. These changes are consistent with those observed in DGE analysis, for which 678 genes were upregulated while 201 were down-regulated. GO analysis of genes in DTE also found that, as with the statistically significant genes in DGE, growth related processes were highly enriched in 6–10 weeks’ gestation, while immune related functions were enriched in 11–23 weeks’ gestation. Transcripts from 861 individual genes were identified using DTE (Figure 2), with 749 expressing only a single DE transcript while 112 genes expressed two or more DE transcripts. The genes with the greatest number of DE transcripts were *NUCB2* and the previously mentioned *CD36*, which each expressed six DE transcripts, representing complex expression patterns within these genes (Appendix A).

The top 10 most significantly upregulated transcripts in the later gestational group, ordered by FDR, were *WNT10A-001*, *MSC-001*, *C8orf4-001*, *DPYSL3-001*, *NALCN-001*, *SVEP-003*, *ANGPTL1-003*, *PREX2-001*, *ALPP-001* and *TMEM176B-004* (Appendix A). The *WNT* gene family transcript *WNT10A-001* and musculin transcript *MSC-001* had the highest logFC (logFC > 3) of all upregulated transcripts. The top 10 most significantly down-regulated transcripts were *HBZ-001*, *HBE1-201*, *CLDN10-002*, *CLDN10-001*, *SLC16A3-018*, *TUBB3-001*, *CGB7-001*, *CLDN6-201*, *GOLT1A-001*, and *C7orf71-001* (Appendix A). Both *HBZ-001* and *HBE1-201* transcripts, members of the hemoglobin subunit family of genes, had the largest changes in expression (|logFC| > 6) of all transcripts, corresponding to the assumed change in oxygen tension between gestational groups. Multiple highly significant transcripts from the claudin gene family, which mediate angiogenesis and trophoblast invasion [32], were found to be down-regulated in mid gestation (*CLDN10-002*, *CLDN10-001*, *CLDN6-201*, *CLDN7-003*, and *CLDN3-001*), with only *CLDN1-001* found to be upregulated.

Significant DTE in the absence of any significant DGE was found for 229 genes. However, when comparing between DGE and DTE analyses, the use of discrete FDR and logFC thresholds may artificially reduce the overlap between DE genes and transcripts. With the thresholds used here (FDR < 0.05, |logFC| > 1), 179 genes were considered unique to DTE, and 247 genes were considered unique to DGE analysis (Figure 2). To identify transcripts that appeared to be genuinely lacking in any significant DGE, but were still significant in DTE analysis, genes with the greatest disparities in gene- and transcript-level results were filtered for by comparing the logFC and FDR from both DGE and DTE. Only genes which were non-significant in DGE (FDR > 0.05) and which showed a discrepancy (>1) between logFC estimates were considered as truly unique to DTE. Under these stringent criteria, only 15 uniquely DE transcripts remained (Figure 3) with no gene-level differential expression (Appendix A). Six of these transcripts (*ADAM10-015*, *VMP1-004*, *MTUS1-005*, *GPR126-009*, *RPS24-001*, *GDPD5-012*, and *TFPI2-002*) were also identified in DTU analysis (FDR < 0.05). Of the 15 transcripts, 7 were upregulated and 8 were down-regulated at 11–23 weeks’ compared to 6–10 weeks’ gestation with four clusters identified (Figure 3). Hierarchical clustering identified two groups which contained only upregulated transcripts which were *ITSN1-001*, *GDPD5-012*, and *MGAT1-004* in the larger cluster and *TFPI2-002* and *FLT1-001* in the smaller cluster. One contained only down-regulated transcripts *PSG9-008*, *HBA2-002*, and *SLC16A3-018*, and one contained the transcripts *CALM1-005*, *MTUS1-005*, *VMP1-004*, *AZIN1-012, ADAM10-015*, *RPS24-001*, and *PSG5-005*, which were DE in either direction. Genes in this set also exhibited transcript expression masking paradigms including collapsing of expression (*FN1*), dominant transcript expression (*PSG9*) (Appendix A) and cancellation of expression (*VMP1*) [17] (Appendix A).

### 2.3. Global Differences in Transcript Usage Were Observed from Early to Mid Gestation

DTU analysis was performed comparing data at 6–10 weeks’ and 11–23 weeks’ gestation to identify genes exhibiting changes in transcript proportions across the 10–11 weeks’ transition period. A total of 2307 genes were analyzed that satisfied the initial filtering criteria, of which 611 were statistically significant for gene-level DTU (Appendix A). Changes in individual transcript proportions between sample groups were quantified by differences between median proportions of transcripts compared between 6–10 weeks’ and 11–23 weeks’ gestation (Figure 4). A total of 582 transcripts were found to decrease in proportion from 6–10 weeks’ to 11–23 weeks’ gestation, while 598 transcripts were shown to increase from 6–10 weeks’ to 11–23 weeks’ gestation. Two patterns of DTU were observed. The first is when two transcripts from the same gene show changes of similar magnitude, but in opposite directions. Whilst this may be expected to indicate a switch between two dominant isoforms, a more common pattern was a large magnitude change in one of the two isoforms, as seen for *ADAM10* (Figure 5).

The most highly ranked genes with DTU were *ADAM10*, *VMP1*, *MTUS1*, *GPR126 (ADGRG6)*, *RPS24*, *GALNT11*, *ASAH1*, *SLK*, *C6orf89*, and *GDPD5* (Appendix A). *ADAM10* had the most significant DTU of all genes, expressing two transcripts (*ADAM10-001*, *ADAM10-015*) with significantly changing proportions (Figure 5). Of the genes exclusively significant in DTU analysis, *ASAH1* was the most highly ranked gene by FDR. Three transcripts (*ASAH1-001, ASAH1-002, ASAH1-006*) were expressed and had significantly changing proportions (Appendix A). A total of 77 DTU genes were also found in DTE analysis, with 27 of those genes also found in DGE analysis (Figure 2) and 50 found in DTE but not in DGE (Appendix A). Four transcripts highlighted in Figure 4 were also featured in the 15 transcripts shown in Figure 3 (*ADAM10-015*, *RPS24-001*, *TFPI2-002* and *VMP1-004*).

The *VMP1* and *GPR126* genes showed the greatest changes in proportion in their transcripts, *VMP1-001*, *VMP1-004*, and *GPR126-009* while *TFPI2-001* and *TFPI2-*002 were the most highly expressed transcripts with statistically significantly changing proportions (Figure 4). *VMP1* exhibited no change in gene expression, yet statistically significant changes were observed in transcript expression. Two transcript isoforms of *VMP1* had statistically significant changes in proportion (FDR < 0.05), including a shorter, protein-coding, transcript with 12 exons (*VMP1-001*), and a longer, non-protein-coding transcript featuring a retained intron (*VMP1-004*) (Appendix A). *GPR126* showed no change in gene expression between 6–10 weeks’ and 11–23 weeks’ gestation despite significant changes at the transcript level. Transcript-level DTU showed *GPR126-009* and *GRP126-003* had the most statistically significant changes in proportions, while *GPR126-009* had the greatest magnitude of change in proportion while *GPR126-003* had a smaller magnitude of change (Appendix A). *TFPI2* exhibited DTU between a non-coding transcript featuring a retained intron (*TFPI2-002*) that was DE in transcripts and increased proportions in 11–23 weeks’ gestation placenta, against a protein coding transcript (*TFPI2-001*) which decreased in proportion (Appendix A). Other transcripts exhibiting both DTE and DTU included *MTUS1-005*, *RPS24-001*, *PSG6-003*, and a processed transcript *ADAM10-015*, while the transcripts *RPS24-003*, *PSG6-008*, *PTPRJ-001*, *PTPRJ-002* were only identified using DTU analysis (Appendix A).

ADAM metalloproteinase domain 10 (*ADAM10*) gene was the most statistically significant in DTU, with large changes in proportion observed for transcripts *ADAM10-001* and *ADAM10-015* (Figure 5). *ADAM10-001* is a longer protein-coding transcript with a larger 3′ UTR region, *ADAM10-015* and *ADAM10-008* are long non-coding transcripts, and *ADAM10-002* is targeted by NMD. *ADAM10* was not considered significant in the DGE analysis, but differential expression was observed for *ADAM10-015* and a subtle decrease in expression was detected in *ADAM10-002* with DTE analysis. The remaining two transcripts showed no change in expression between gestational windows. While no differential expression was identified for the *ADAM10* gene in the DGE analysis, an overall decrease was observed which mirrored the decrease in *ADAM10-15*. Importantly, whilst the proportion of *ADAM10-001* appeared to change, the overall levels were constant indicating this putative change was an artefact primarily due to the decrease in *ADAM10-015*.

## 3. Discussion

To our knowledge, this study offers the first profile of transcript expression and usage in placenta across early to mid gestation (6–23 weeks’). Sequencing of a large number of samples has allowed detailed characterization of the normal developing placental transcriptome, while highlighting differential usage of specific transcripts. Here we demonstrate the dynamic and complex transcriptional activity occurring in the human placenta during the transition from a low, but physiological for this time in gestation, oxygen environment (6–10 weeks’) to a more “normal” oxygen (11–23 weeks’) environment through DGE, DTE and DTU analyses. The transition of oxygen tension from 6–10 weeks’ to 11–23 weeks’ gestation was marked by distinct changes in transcript-level dynamics and enriched for pathways including cell migration, transmembrane signalling receptor activity, cellular responses to chemical stimulus and organic substances, and defense response.

Integrating DGE, DTE, and DTU analysis results provided a comprehensive representation of the placental transcriptome and highlighted changes in transcript expression and usage in the absence of any change in gene expression. As a result, DE genes with significant changes in expression and proportion in multiple transcripts were observed. For example, *CD36*, a gene upregulated in hypoxia that possesses a HIF-1 binding site [33] and is involved in mediation of angiogenesis and inflammatory response [34] had 6 DE transcripts and one transcript with changing proportions. *NRP2*, which encodes the neuropilin-2 receptor that is repressed by hypoxia and regulates both VEGF and SEMA3F activity to induce tumor angiogenesis [35], also exhibited DGE with two DE transcripts and one transcript with changing proportions. Both *CD36* and *NRP2* had only one transcript with significant proportion changes each but proportion changes cannot theoretically occur with only one transcript. Hence these changes were found to be the result of subtle non-significant changes in the proportion of transcripts from the same gene in the opposite direction of the significant transcript (Appendix A).

Profiling at an individual transcript resolution afforded an overview of transcript configuration within genes. The *FLT1* gene is well known to express a soluble isoform (*sFLT-1*) that can tightly bind *VEGF* and suppress angiogenic activity [31] and is associated with preeclampsia [36]. Multiple isoforms of *FLT1*, including a soluble isoform, have been identified in this study through transcript level profiling. The *FLT1-201* isoform exhibits alternative splicing at exon 12, resulting in a shorter transcript, while *FLT1-203* deviates from *FLT1-001* at an alternatively spliced exon 15a to form a 733aa soluble Flt1 (sFlt1-e15a) isoform identified in a previous study [31] which is significantly upregulated in preeclampsia [37,38]. Statistically significant changes in the expression of non-coding transcripts previously associated with cell invasion, migration, and proliferation, were also identified. *MSC-AS1*, an anti-sense lncRNA, had higher expression at 11–23 weeks’ gestation and has previously been found to enhance proliferation of gastric cancer cells [39]. Higher expression of *C1QTNF1-AS1* was identified at 6–10 weeks’ gestation and overexpression of this lncRNA has been shown to impede proliferation, migration and invasion of human hepatocellular carcinoma cells [40].

It is becoming increasingly apparent that the expression of the majority of all genes is driven by expression of the dominant transcript [41] and so a greater overlap between DGE and DTE genes represented in Figure 2 would have been expected. Comparison of discrepancies between DGE and DTE found variable expression patterns between transcripts within a gene appeared to mask detection of significant changes in overall gene expression. Masking occurred through the collapse of similar transcript-level expression patterns to the gene-level, cancellation of transcripts changing in opposing directions, and the presence of a dominant transcript [17]. The top 15 transcripts with the greatest disparities compared to the gene-level highlighted the consequential masking of transcript variability upon aggregation to an overall gene expression value. Dominant expression masked minor DE transcripts in *HBA2*, *MGAT1,* and *PSG9,* while collapse of subtle changes in expression of multiple transcripts to the gene-level masked significant DTE in *GDPD5*, *AZIN1*, and *PSG5*. A switch in dominant isoform was also observed in *VMP1* (Appendix A). These examples suggest that many genes with complex transcript expression patterns cannot be identified upon gene-level aggregation. It has also been previously reported that variable transcripts escape identification in DGE analyses, as the presence of alternative isoform usage leads to an inflated FDR [22].

The interpretation of transcript proportions in DTU analysis presents an opportunity for important insights into the biology but opens the door to possible misinterpretation. Whilst it commonly appears that two isoforms are being switched, it is instead likely to be one isoform being up- or down-regulated as required while the remaining isoforms are unchanged. This is potentially a key insight into the underlying biology that is commonly missed by differential expression analysis at the gene-level [28,42]. Increased transcript diversity per gene is possible through modifications by alternative splicing and alternative promoter usage. These modifications can also target a transcript for nonsense-mediated-decay (NMD) or loss of protein coding capability through intron retention [43]. Specific transcript isoforms have been identified herein that are potential candidates for eliciting functional consequences on the development of the human placenta in early gestation, including both protein-coding and non-coding transcripts. Genes with significant DTE also showed changing transcript proportions which were only detectable through sensitive DTU methods.

High enrichment of cell signaling, cell migration, and immune related pathways were identified from genes statistically significant in DGE, DTE and DTU analyses but only two significantly enriched pathways were oxygen related. However, acute responses to hypoxia occur during early placental development (<10 weeks’ gestation) that are mediated by HIF-1α which regulates developmental processes including trophoblast proliferation in villus explant tissues from first trimester placenta [44]. Despite an absence of enrichment of any hypoxia related processes, genes with the highest significance in DTU (FDR < 7.3 × 10^−26^) including *ADAM10*, *VMP1*, and *GPR126* are known to be associated with hypoxia responsive processes such as angiogenesis and autophagy.

*ADAM10* encodes disintegrin and metalloproteinase domain 10 and is upstream of the *NOTCH* and *VEGF* signaling pathways, of which the *VEGF* pathway is associated with angiogenesis [45]. ADAM10 has also been shown to mediate the release of the soluble Flt-1 isoform, a known marker of preeclampsia [46] and knockdown of *ADAM10* leads to decreased sFlt-1 [47,48]. Even though no change in gene expression was observed, significant DTE and DTU was found in *ADAM10* transcripts. Reduced expression and proportion of the non-coding *ADAM10-015* transcript occurred simultaneously with an increase in proportion of the protein coding *ADAM10-001* transcript which contains the functional disintegrin and peptidase domains [49]. Although the function of *ADAM10-015* is unknown, a 10-fold decrease in expression across early gestation may suggest hypoxia responsiveness in that transcript (Figure 5). In fact, increased expression of *ADAM10* has previously been shown to be mechanistically linked to hypoxia-induced accumulation of HIF-1α [50]. The *ADAM10-002* transcript which was the third most abundant of *ADAM10* transcripts is known to undergo NMD, potentially indicating a regulatory role of alternative splicing by targeting transcripts for NMD [51,52].

Changes in transcript expression were observed in *GPR126* while gene-level changes were undetected. DTU was observed for *GPR126-008*, *GPR126-009*, and *GPR126-003* with the greatest changes in proportions occurring in *GPR126-009* and *GPR126-003*, which differ by an 84-nucleotide exon. *GPR126-008* and *GPR126-009* each possessed a unique exon (Appendix A). Interestingly, a skipping event of exon 23 in *GPR126* has been previously shown to be associated with Intrauterine growth restriction (IUGR) and angiogenic-related processes in the human placenta [25]. Interestingly, upon comparison of exon coordinates, it was found that the exon uniquely encoding the *GPR126-008* transcript matched the skipped exon associated with IUGR. Knockdown of *GPR126* has also been found to inhibit hypoxia-induced angiogenesis in mouse retina [53]. *GPR126-009* was more highly expressed in placenta from 11–23 weeks’ compared to 6–10 weeks’ gestation while both *GPR126-003* and *GPR126-008* significantly decreased in proportions. These suggest potential opposing modes of regulation for each individual transcript in response to hypoxia.

Vacuole membrane protein 1 (*VMP1*) is a gene known to be involved in autophagy. A steady increase in proportion and expression of the non-coding *VMP-004* transcript that features a retained intron was observed, where it replaced the protein-coding *VMP1-001* as the dominant transcript around 9–11 weeks’ gestation. Intron retention is mediated through multiple levels of regulation and in mature transcripts it may serve to either alter the resulting protein or cause the transcript to completely lose protein coding potential [54]. HIF-1α induced autophagy has been shown to be reduced with down-regulation of *VMP1* in human colon cancer cell lines [55]. A hypoxia signature miRNA, miR-210, has also been shown to down-regulate *VMP1* expression and consequently promote cell migration and invasion [56]. Down-regulation of *VMP1-004* was observed in early (6–10 weeks’) gestation (Appendix A), coincident with hypoxic conditions in which increased levels of hypoxia-induced autophagy, miR-210 expression, cell migration and invasion have been reported [44,57].

The *ASAH1* gene expressed three transcript variants that result in differences in the protein active site upon translation. Uniprot annotation [58] of *ASAH1-001* showed that it possesses four sites, Cys143, Arg162, Asn320, and Arg333, necessary for the activation of acid ceramidase [59], while *ASAH1-006* contains a retained intron that prevents protein translation. Interestingly, a study in human melanoma cells showed lysosomal acid ceramidase controls the transition between invasive and proliferative phenotypes, with reduced *ASAH1* expression associated with invasive behavior [60]. *ASAH1* showed a subtle non-significant increase (|logFC| > 0.4 and FDR = 1) in *ASAH1-001* and a significant decrease (|logFC| > 0.9 and FDR < 0.05) in *ASAH1-006* expression that was filtered out by the effect size cut-off (|logFC| > 1). However, a significant (FDR < 0.05) increase in *ASAH1-001* and decrease in *ASAH1-006* proportion was detected in DTU. Relatively lower *ASAH1-001* and higher *ASAH1-006* proportions were found at 6–10 weeks’ gestation, coinciding with hypoxic conditions and trophoblast invasion in the first trimester [1,3].

In conclusion, this study is the first to profile variable human placental expression and usage of transcripts across early to mid gestation by sequencing samples from a large number of pregnancies. Overall, profiling transcriptional dynamics at an individual transcript resolution captured changes between 6–10 weeks’ and 11–23 weeks’ gestation. These appear to reflect the dynamic changes occurring in placental differentiation and growth as it transitions from a physiologically low oxygen environment to a more “normal“ oxygen environment. Analysis of dynamic placental transcriptional change in early to mid gestation in uncomplicated pregnancies is essential before we can identify aberrant transcription that underpins later pregnancy complications. The next step is to determine functional roles of specific gene isoforms in early human placental development in normal and pathological pregnancy.

## 4. Materials and Methods

### 4.1. Data Information and Ethics Statement

The RNA-sequencing data used in this study was sourced from previously sequenced human placental chorionic villus samples from 6–23 weeks’ gestation reported in Breen et al. [10]. All RNA-sequencing data is available for download at NCBI Gene Omnibus (GEO) under the Accession number GSE150830. Ethical approval was obtained from the Queen Elizabeth Hospital Human Research Ethics Committee (HREC/16/TQEH/33; 10 May 2016).

### 4.2. Data Processing

The data were reanalyzed using selective alignment for identification of individual transcript expression profiles. Initial FastQC reports were generated from raw FASTQ files with *FastQC* [61] and visualized through the *ngsReports* R package [62], underwent trimming of sequencing adapters by *AdapterRemoval* [63] and then selective alignment using the *Salmon* package (v1.1.0) by specifying the—*validateMappings* argument during alignment [64]. The selective alignment index was generated using the Ensembl GRCh37 build of the reference human transcriptome [65] with the Gencode v19 GRCh37 reference annotation [66]. As sequence fragments from unannotated regions of the transcriptome may falsely align to annotated transcripts bearing sequence similarity, the index was augmented with decoy transcript sequences to be used during selective alignment to reduce false mappings [67]. Quantification of transcript counts was performed with the default *Salmon quant* method. Transcript counts were divided by bootstrapping estimates of transcript overdispersion to reduce mapping uncertainty arising from physical overlap of transcript regions, as suggested in the *edgeR* reference manual [68]. Gene counts were generated by aggregating the raw output of transcript counts from *Salmon* to the gene-level and were used for DGE analysis. Due to unequal male and female placental sample numbers and high variability of transcript expression, transcripts from the X and Y chromosomes were removed prior to performing any further analyses.

### 4.3. Differential Expression Analysis

Both gene and transcript expression analyses were performed on counts filtered for >2 CPM in ≥27 samples, with 27 samples representing the smallest sample group in the comparison between 6–10 weeks’ (*n* = 27) and 11–23 weeks’ (*n* = 57) gestation. A comparison between global distributions of reads from the two sample groups was performed using *quantro* [69]. Differences between read distributions were detected (*p* < 0.05), prompting the use of smooth quantile normalization through *qsmooth* [70]. DGE and DTE analyses were performed using a generalized linear fit model (*glmQLFit*) implemented in *edgeR* [68,71], comparing 6–10 weeks’ and 11–23 weeks’ gestation, with fetal sex as a covariate. The gene-wise statistical tests were performed relative to a specified log fold-change threshold of 1.4 using *glmTreat*. Significance of DGE and DTE was measured at an FDR < 0.05 and |logFC| > 1. Transcripts in the heatmap were clustered using the Ward’s minimum variance method with dissimilarities squared before clustering (“ward.D2”) [72] and visualized via *pheatmap* [73]. The DGE analysis performed in this study is a reanalysis of RNA-seq data from a study by Breen et al. (2020), while implementing smooth quantile normalization and the same differential expression testing methods described above to remain consistent with the DTE analysis.

### 4.4. Differential Transcript Usage

Normalized counts were converted to transcripts per million (TPM) scaled by transcript length using *tximport* [22], then filtered for transcripts contributing at least 10% of the respective gene expression with >2 read counts in ≥27 samples as above. Using *DRIMSeq* [74], samples were then tested for DTU [23] using a Dirichlet-multinomial model to test at the gene-level and beta-binomial model to test at the transcript-level with significance of DTU measured at an FDR < 0.05, prior to validation of DTU with *stageR* [75]. To quantify the magnitude of change in transcript proportions for visualization, the median proportions of each transcript within each sample group, 6–10 weeks’ and 11–23 weeks’ gestation, were calculated. The difference in median proportions between each group was then used to represent the magnitude of proportion change.

### 4.5. Gene Ontology Enrichment

As GO annotations are only provided comprehensively at the gene-level, gene identifiers were used for enrichment testing for DGE result and both DTE and DTU results. GO enrichment was performed using *goseq* [76] allowing identification of biases in selection of DGE/DTE/DTU genes through the *nullp* function. Gene length, transcript length, GC content, and median transcript length per gene information were sourced from *ensembldb* [77] and used to test for bias. The median transcript length per gene accounted for the greatest bias for DTE and DTU results while gene length was the primary bias in the DGE genes. The Wallenius method was used to determine GO enrichment, accounting for biases, and the resulting over-representative *p*-values were adjusted via the Benjamini-Hochberg method for false discovery [78].

## Figures and Tables

**Figure 1 ijms-23-04506-f001:**
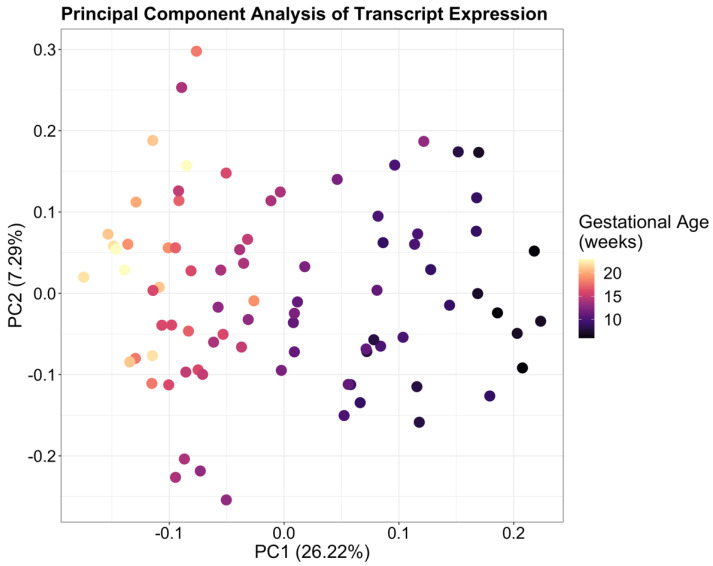
Principal Component Analysis of transcript-level expression in placenta showing a clear association between gestational age across 6–23 weeks’ gestation and PC1. Samples in the earlier weeks of gestation are localized on the right with mid-gestation samples on the left, indicating that gestational age represents the largest source of variability within the data.

**Figure 2 ijms-23-04506-f002:**
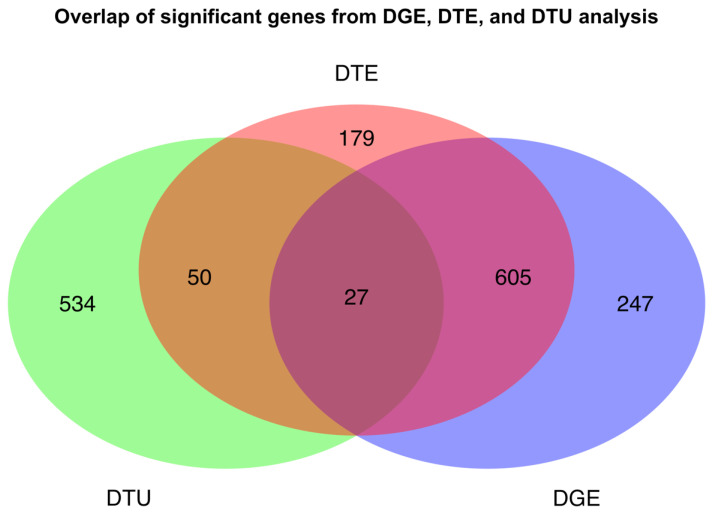
Overlap of significant genes from DGE, DTE, and DTU analyses between 6–10 weeks’ and 11–23 weeks’ gestation placenta. Statistical significance was determined using an FDR < 0.05 and |logFC| > 1 for both DGE and DTE analysis, and an FDR < 0.05 for DTU.

**Figure 3 ijms-23-04506-f003:**
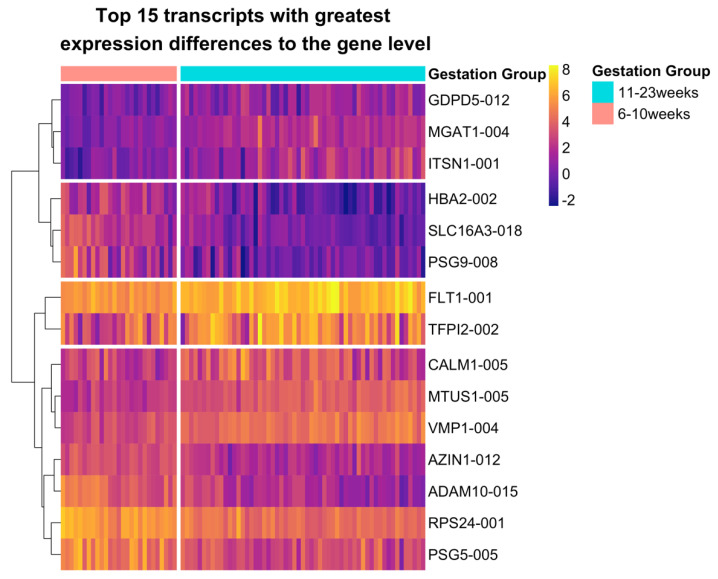
Expression of the top 15 transcripts in DTE analysis that exhibited the greatest difference in expression compared to DGE analysis. Rows (transcripts) are horizontally clustered using the Ward D2 method with gaps distinguishing each cluster, and columns (samples) are ordered by gestational age. A vertical gap in the heatmap separates the counts into the two sample groups at 6–10 weeks’ and 11–23 weeks’ gestation.

**Figure 4 ijms-23-04506-f004:**
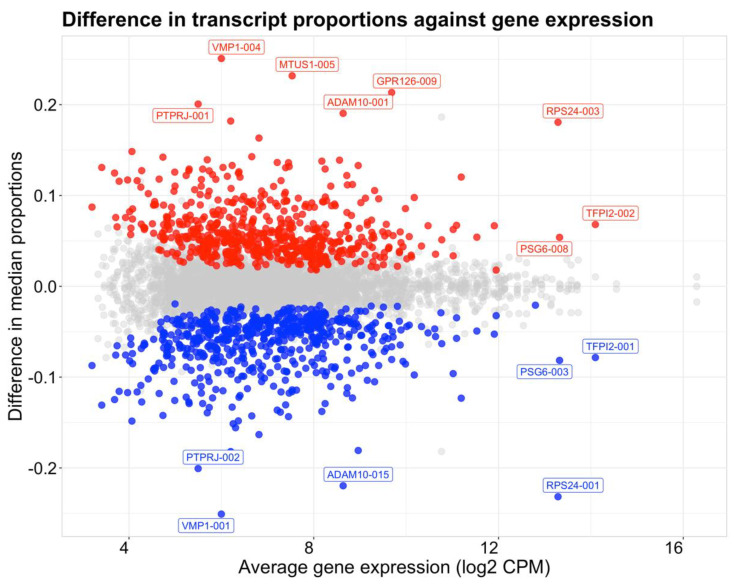
Scatterplot showing median transcript proportion differences compared to average gene expression (log2 CPM) between placental villus tissue at 11–23 weeks’ and 6–10 weeks’ gestation. Significant transcripts (FDR < 0.05) with a difference in median proportions between sample groups outside the interquartile range were highlighted and colored by the direction of change. Red points indicate an increase in proportion of a transcript from 6–10 weeks’ to 11–23 weeks’ gestation and blue points indicate a decrease from 6–10 to 11–23 gestational weeks. Transcripts with the largest changes in proportion (>0.2) or changing transcripts with the highest average gene expression (>12) are labelled.

**Figure 5 ijms-23-04506-f005:**
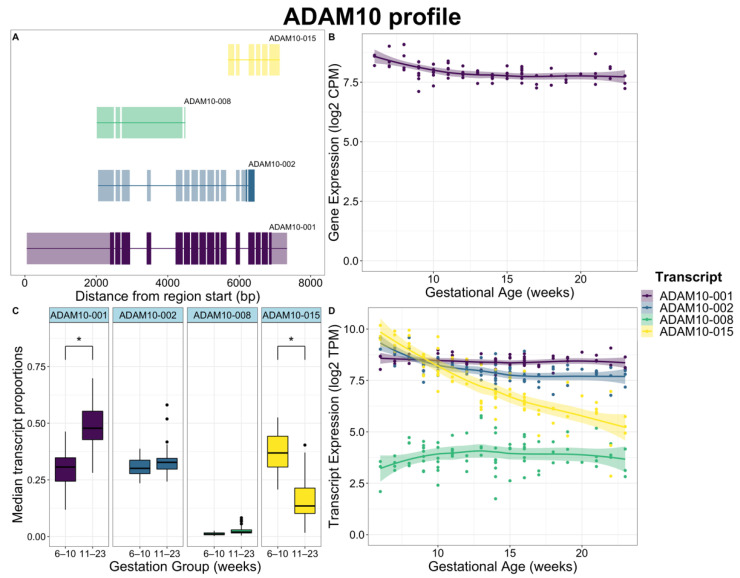
Placental villus *ADAM10* gene expression, transcript expression, and transcript usage between 6–10 weeks’ and 11–23 weeks’ gestation with intron-exon structures of each transcript. (**A**) Structures of *ADAM10* isoforms. Coding sequences in each transcript are opaque with non-coding regions transparent. All transcripts are transcribed on the reverse strand. Each transcript is positioned relative to the gene region start site. (**B**) Gene expression of *ADAM10* in log2 CPM across 6–23 weeks’ gestation. (**C**) Boxplot of *ADAM10* transcript proportions of total gene expression between 6–10 weeks’ and 11–23 weeks’ gestation. Transcripts with significant changing proportions are indicated (*). (**D**) Transcript expression in log2 TPM scaled by transcript length of *ADAM10* across 6–23 weeks’ gestation, showing a near 32-fold (log2 5-fold) decrease in expression of the non-coding transcript *ADAM10-015*.

**Table 1 ijms-23-04506-t001:** Top 20 enriched GO terms from gene ontology analysis using the 1642 genes that were significant in DGE, DTE, and DTU.

Category	Term	Ontology	* DE in Term	FDR
GO:0005886	plasma MEMBRANE	CC	637 (21.0%)	$8.22\times 10^{-38}$
GO:0031226	intrinsic component of plasma MEMBRANE	CC	237 (28.4%)	$1.19\times 10^{-28}$
GO:0005887	integral component of plasma MEMBRANE	CC	228 (28.9%)	1.34 × 10^−28^
GO:0016021	integral component of MEMBRANE	CC	559 (19.9%)	$6.94\times 10^{-25}$
GO:0004888	transmembrane signaling receptor ACTIVITY	MF	137 (31.6%)	$2.95\times 10^{-20}$
GO:0005102	signaling receptor BINDING	MF	220 (24.7%)	$1.81\times 10^{-18}$
GO:0016477	cell MIGRATION	BP	238 (23.8%)	6.55 × 10^−18^
GO:0070887	CELLULAR RESPONSE to chemical stimulus	BP	414 (20.0%)	$9.22\times 10^{-18}$
GO:0006952	defense response	BP	232 (23.7%)	3.52 × 10^−17^
GO:0071310	CELLULAR RESPONSE to organic substance	BP	342 (20.7%)	3.29 × 10^−16^
GO:0005539	glycosaminoglycan BINDING	MF	59 (44.7%)	$6.87\times 10^{-16}$
GO:0030334	REGULATION of cell MIGRATION	BP	168 (25.8%)	$1.42\times 10^{-15}$
GO:0048018	receptor ligand ACTIVITY	MF	68 (39.8%)	$2.83\times 10^{-15}$
GO:0007166	cell surface receptor signaling pathway	BP	355 (20.1%)	$4.72\times 10^{-15}$
GO:0051270	REGULATION of cellular component movement	BP	180 (24.8%)	$5.55\times 10^{-15}$
GO:2000145	REGULATION of cell motility	BP	171 (25.1%)	$8.59\times 10^{-15}$
GO:0030546	signaling receptor activator ACTIVITY	MF	69 (38.5%)	$9.19\times 10^{-15}$
GO:0008201	heparin BINDING	MF	48 (49.0%)	$9.70\times 10^{-15}$
GO:0006935	chemotaxis	BP	109 (29.9%)	$2.19\times 10^{-14}$
GO:0031982	vesicle	CC	509 (18.1%)	$9.87\times 10^{-14}$

* The number of genes in each GO term category that were statistically significant are shown in “DE in category” with the percentage showing the proportion of DE genes in the total genes within a category. The “Ontology” column highlights whether the term is a biological process (BP), molecular function (MF), or a cellular component (CC). Capitalized words within terms indicate the presence of a parent or child term within the table. Two terms with the same capitalized words (i.e., “REGULATION of cell MIGRATION” and “REGULATION of cell motility” or “REGULATION of cell MIGRATION” and “cell MIGRATION”) indicate these terms have a shared ancestry.

## Data Availability

The full workflow for this study is available at (https://github.com/JBogias/EarlyGestationPlacentaProfile). All sequencing data is available for download at NCBI Gene Expression Omnibus (GEO) under the Accession number: GSE150830.

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
