# Peer review of "Placental Transcription Profiling in 6–23 Weeks’ Gestation Reveals Differential Transcript Usage in Early Development"

_ijms, 2022, doi:10.3390/ijms23094506_

Round 1
Reviewer 1 Report
Transcript expression profiles were obtained for 84 human chorionic villus samples across 6–23 weeks gestation.
These samples were analyzed for differential gene expression (DGE), differential transcript expression (DTE), and differential transcript usage (DTU). Many genes exhibited DTE without DGE (that is the expression of the gene did not change but the transcripts being used changed over the course of development). Even more genes had DTU without DGE or DTE. The difference between DTU and DTE is that DTU measures changes in the proportion of transcripts being used over time. Highly significant DTU genes were associated with hypoxia-responsive pathways, probably associated with increased oxygenation of the intervillous space as early pregnancy progresses.
I was intrigued to see a shift in the DTU data for TFPI2 between a coding and non-coding transcript and wondered whether this might have something to say about the variable placenta-specific imprinting reported for this gene (Genome Research 18: 1270).
This is the report and high-level analysis of what will be a very useful dataset.
Reviewer 2 Report
Human placental development is unique in that sense that it is the most rapidly developing transient organ, and this type of development, which resembles much to that of a tumor, is key in the success of pregnancy. To uncover this unique developmental process, the authors investigated the characteristic dynamic changes in the placental transcriptome between 6-23 weeks of gestation in a large number of placental specimens. Differential gene and transcript expression as well as differential transcript usage were assessed in two time-periods, before and after the placental circulation is established. This remarkable change was well observed in the most significant differential transcript usage in genes involved in hypoxia-responsive pathways. Along with other changes, these observations gave a very good view into the regulatory mechanisms switching on in early placentation, which can also be the basis for the further understanding of the origins of pregnancy complications.
This is a very well designed and executed study. The results are of importance, clearly presented and discussed. The study is well referenced and all the necessary insights are provided how these novel data is fitting the known literature. This reviewer enjoyed reading the paper and has only very minor issues to request to be changed before the paper can formally be accepted for publication.
- The format of some references do not seem to be entirely correct (e.g. Refs 6, 35). Please double check all and correct where needed.
- Reference 10 publication record is not provided. Please provide.
